# Single Nucleotide Polymorphism in the *IL17A* Gene Is Associated with Interstitial Lung Disease Positive to Anti-Jo1 Antisynthetase Autoantibodies

**DOI:** 10.3390/life11020174

**Published:** 2021-02-23

**Authors:** Marco Antonio Ponce-Gallegos, Montserrat I. González-Pérez, Mayra Mejía, Karol J. Nava-Quiroz, Gloria Pérez-Rubio, Ivette Buendía-Roldán, Espiridión Ramos-Martínez, Jorge Rojas-Serrano, Ramcés Falfán-Valencia

**Affiliations:** 1HLA Laboratory, Instituto Nacional de Enfermedades Respiratorias Ismael Cosio Villegas, Calzada de Tlalpan 4502, Sección XVI, Tlalpan, México City 14080, Mexico; marcoapg@iner.gob.mx (M.A.P.-G.); knava@iner.gob.mx (K.J.N.-Q.); glofos@yahoo.com.mx (G.P.-R.); 2Interstitial Lung Disease and Rheumatology Unit, Instituto Nacional de Enfermedades Respiratorias, Ismael Cosio Villegas, Calzada de Tlalpan 4502, Sección XVI, Tlalpan, México City 14080, Mexico; ixchelglez19@gmail.com (M.I.G.-P.); medithmejia1965@gmail.com (M.M.); 3Translational Research Laboratory on Aging and Pulmonary Fibrosis, Instituto Nacional de Enfermedades Respiratorias Ismael Cosio Villegas, Calzada de Tlalpan 4502, Sección XVI, Tlalpan, Mexico City 14080, Mexico; ivettebu@yahoo.com.mx; 4Experimental Medicine Research Unit, Facultad de Medicina, Universidad Nacional Autónoma de México, Mexico City 06720, Mexico; espiri77mx@facmed.unam.mx

**Keywords:** *IL17A*, anti-tRNA, SNPs, ASSD, anti-Jo1

## Abstract

Antisynthetase syndrome (ASSD) is a rare multisystemic connective tissue disease affecting the skin, joints, muscles, and lungs, characterized by anti-aminoacyl transfer-RNA-synthetases (anti-tRNA) autoantibodies production, being anti-Jo1 the most frequent. We included one-hundred twenty-one ASSD patients and 340 healthy subjects (HS), and also, we divided the case group into anti-Jo1 and non-anti-Jo1. Two single nucleotide polymorphisms (SNPs) in the *IL17A* gene were evaluated. Anti-Jo1 was the most common anti-tRNA antibody in our cohort, and the most frequent tomographic pattern was non-specific interstitial pneumonia (NSIP). Anti-Jo1 ASSD patients had higher levels of creatine phosphokinase than the non-anti-Jo1 group. Significant differences in genotype frequencies with rs8193036/CC between anti-Jo1 vs. non-anti-Jo1 ASSD patients (*p* < 0.001), maintaining the association after Bonferroni correction (*p* = 0.002). Additionally, in the anti-Jo1 group vs. HS comparison, we found a statistically significant difference with the same SNP (*p* = 0.018, OR = 2.91, 95% CI = 1.15–7.35), maintaining the association after Bonferroni correction (*p* = 0.036). The rs8193036/CC genotype in *IL17A* is associated with ASSD patients with anti-Jo1. Also, anti-Jo1 and non-anti-Jo1 patients display differences in genotype frequencies.

## 1. Introduction

Antisynthetase syndrome (ASSD) is defined as a rare multisystemic connective tissue disease, characterized by affects predominantly the skin, joints, muscles, and lungs [1], as well as the presence of the anti-aminoacyl transfer-RNA-synthetases (anti-tRNA) autoantibodies, which include anti-Jo1 (anti-histidyl), anti-PL12 (anti-alanyl), anti-PL7 (anti-threonyl), anti-EJ (anti-glycyl), anti-OJ (anti-isoleucyl), anti-SC (anti-lysil), anti-KS (anti-asparaginyl), anti-JS (anti-glutaminyl), anti-Ha (anti-tyrosyl) or anti-YRS (anti-threonyl), anti-tryptophanyl, and anti-Zo (anti-phenylalanyl), being anti-Jo1 the most frequent, with a frequency of 68% in patients with ASSD, and 25% in patients with idiopathic inflammatory myopathies (IIM) [2,3,4].

Traditionally, the ASSD was firstly described in association with IIM, such as dermatomyositis (DM) and polymyositis (PM) [1,5]. However, nowadays, it seems that these patients can only present interstitial lung disease (ILD) and anti-tRNA autoantibodies without fulfilling IIM classification criteria [6,7]. Additionally to anti-tRNA autoantibodies, patients with IIM present diverse myositis associated antibodies, being Ro52 the most prevalent. In this way, previous studies have described that Ro52 is an independent risk factor for IIM-related ILD [8]. Among the principal ASSD clinical manifestations, ILD is the most frequent and severe, with an incidence of approximately 80–90% [9,10].

Due to the heterogenicity of the ASSD, little is known about its pathophysiology and its genetic background. Few previous reports have shown a relationship between anti-tRNA autoantibodies and the human leukocyte antigen (HLA) loci, specifically with the HLA-DRB1*08:03 allele in the Korean population. Also, HLA-DRB1*12:02 and HLA-DRB1*14:03 were associated with DM and PM, respectively [11]. Also, Chinoy et al. [12] described that in PM/DM combined, HLA-DRB1*03, DQA1*05, and DQB1*02 were all influential risk factors for the presence of anti-tRNA synthetase autoantibodies. Another important gene that has been widely studied in diverse autoimmune diseases is the *IL17A* gene, which has been described as an important genetic risk factor. For example, there are studies of single nucleotide polymorphisms (SNPs) in *IL17A* associated with rheumatoid arthritis (RA) and systemic lupus erythematosus (SLE) [13,14,15,16]. However, there are no previous reports about the relationship between ASSD and genetic variants in *IL17A*. Interestingly, our research group recently described an association between cytokines of the Th17 inflammatory profile and ASSD patients who presented disease progression [4], suggesting an essential role of the Th17 related cytokines in the disease pathogenesis.

For all the above described, we aimed to evaluate two promoter SNPs in the *IL17A* gene in the genetic susceptibility for ASSD and the various anti-tRNA autoantibodies.

## 2. Materials and Methods

### 2.1. Subjects Included

#### 2.1.1. Cases Groups

We included 121 patients with ASSD diagnosis in this case-control study. All of them were evaluated and managed in the Interstitial Lung Disease and Rheumatology Unit (ILD&RU) at the Instituto Nacional de Enfermedades Respiratorias Ismael Cosio Villegas (INER) in Mexico City, Mexico. In this reference center, patients are evaluated by a multidisciplinary group (pulmonologists, radiologists, and rheumatologists). We included patients ≥18 years old, with the diagnosis of ILD confirmed by high resolution computed tomography (HRCT) and being positive to at least one of the aminoacyl-tRNA synthetase autoantibodies: Jo1, PL7, PL12, EJ, or OJ, as well as Ro52, measured by EUROLINE: Myositis Profile 3 immunoblot 16 strips (EUROIMMUN AG, Lübeck, Germany) according to the manufacturer’s instructions; also, in the Appendix A, a brief test principle description is included. In the Appendix A, we included two immunoblots’ examples. In addition, we included baseline data of pulmonary function obtained from the patient’s clinical records of carbon monoxide diffusing capacity (DLco) and forced vital capacity (FVC). Furthermore, baseline serum creatinine phosphokinase (CPK) levels were recorded, as well as the clinical characteristics, such as Raynaud’s phenomenon, arthritis, mechanic hands, fever, and smoking history. Patients were evaluated between January 2008 to January 2019. The case group was divided into anti-Jo1 and non-anti-Jo1 patients for further analysis and ASSD patients positive and negative to Ro52.

#### 2.1.2. Control Group

A group of three hundred and forty-six healthy volunteer subjects (HS) was also included. These subjects were recruited from INER’s blood bank as healthy subjects. They had the following characteristics: clinically healthy (with neither chronic nor acute self-reported diseases), ≥18 years old, men and women, born as Mexican-Mestizos (MM, parents, and grandparents born in Mexico, not biologically related among themselves or with the patients), and no history of family pulmonary and/or inflammatory/autoimmune diseases. All participants underwent a background questionnaire of demographic and pathological personal history. The subjects who did not meet the inclusion criteria were excluded from this study.

### 2.2. Ethics Approval and Informed Consent

This study was approved by the Institutional Committees for Research, Ethics in Research, and Biosecurity of the INER (approval code numbers: C08-17, B11-19). All participants were previously invited to participate in the study; they signed a written informed consent document and provided a privacy statement describing personal data protection.

### 2.3. DNA Extraction

Firstly, we obtained 15 mL of peripheral blood via venipuncture in two EDTA (ethylenediaminetetraacetic acid) tubes from all 472 subjects. After that, the DNA extraction was performed using the commercial BDtract Genomic DNA isolation kit (Maxim Biotech, San Francisco, CA, USA). Next, the DNA was quantified by UV absorption spectrophotometry at the 260-nm wavelength employing a NanoDrop 2000 device (Thermo Scientific, Wilmington, DE, USA). Contamination with organic compounds and proteins was determined by measuring the ratio absorbance at 260/280. Samples were considered of good quality when the ratio was ~1.8. All samples were adjusted to 50 µg/µL for subsequent genotyping.

### 2.4. SNP Selection

Two SNPs (rs8193036 and rs2275913) in the *IL17A* gene were selected based on a bibliographic search in PubMed (NCBI), identifying polymorphisms previously associated with other inflammatory and respiratory diseases, such as RA, SLE, ILD, and psoriasis. Additionally, these two SNPs promote an increase of cytokine expression, and both SNPs have a minor allelic frequency (MAF) higher than 5% in the Mexican population in Los Angeles, according to the 1000 genomes project [17]. In a previous study, we described the principal characteristics of the *IL17A* gene and the two SNPs selected [18].

### 2.5. SNP Genotyping

The allele discrimination was performed using commercial TaqMan probes (Applied Biosystems, San Francisco, CA, USA), employing qPCR in a 7300 Real-Time PCR System (Applied Biosystems/Thermo Fisher Scientific Inc., Singapore), and the analysis performed by sequence detection software version 1.4 software (Applied Biosystems, CA, USA). Further, three controls without template (contamination controls) were included for each genotyping plate, and 5% of the genotyped in duplicate as controls for allele assignment.

All experiments were performed following the relevant guidelines and regulations. The STREGA (Strengthening the Reporting of Genetic Association) guidelines were considered to design this genetic association study [19].

### 2.6. Hardy-Weinberg Equilibrium and Haplotypes

The Hardyg–Weinberg equilibrium (HWE) was assessed using SNPStats (http://bioinfo.iconcologia.net/SNPstats (accessed on 15 February 2021)) [20]. The haplotype analysis was performed using Haploview software version 4.2 [21], using the criteria established by Gabriel et al. [22].

### 2.7. Statistical Analysis

Clinical quantitative variables were analyzed using SPSS for Windows, v20.0 (SPSS software, IBM, Chicago, IL, USA). Kolmogorov–Smirnov normality test was carried out, and according to these, parametric or non-parametric tests were used as appropriate. Genotype analysis and genetic association models were carried out with Pearson’s chi-squared and Fisher’s exact tests using Epi Info v. 7.1 software (Atlanta, GA, USA) [23], and 2 × 2 contingency tables were made to estimate the genetic association for ASSD. A *p*-value of less than 0.05 was considered statistically significant. To adjust the significance values, the Bonferroni correction (multiple testing) was applied to the association results. Comparisons were made between ASSD and HS, an intra-case analysis, dividing the ASSD patients into anti-Jo1 versus non-anti-Jo1 patients, and anti-Jo1 patients versus HS, as well as ASSD patients, Ro52+ versus non-Ro52, and Ro52+ versus HS.

## 3. Results

### 3.1. Demographic Variables in Case and Control Groups

One hundred and twenty-six patients with ASSD diagnoses were included in the study, and 346 HS as the control group. We did not find any difference between the ASSD group and HS in age, sex, and body mass index (BMI).

Furthermore, 38.46% of the ASSD group are smokers with around 5 packs/year, a median of 20 years of smoking, and five cigarettes per day (data not shown). The most frequent clinical manifestation was arthritis (73.33%), followed by the mechanic’s hands, fever, and Raynaud’s phenomenon. Further, the most frequent ARS autoantibody was anti-Jo1 (42.98%), and the most frequent HRCT pattern was non-specific interstitial pneumonia (NSIP) (43.27%). The complete results are shown in Table 1.

### 3.2. Demographic Variables in Anti-Jo1 and Non-Anti-Jo1 Groups

We divided the case group into anti-Jo1 and non-anti-Jo1, comparing them to each other. Fifty-two patients were included in the anti-Jo1 group, while 69 were non-anti-Jo1. We did not find statistically significant differences between both groups in age, sex, and BMI, as well as smoking status, pulmonary function pre-bronchodilator (FVC-pb) and single-breath carbon monoxide diffusing capacity (DLco), and clinical manifestations (arthritis, mechanic’s hands, Raynaud’s phenomenon and fever, *p* > 0.05). Furthermore, there were no significant differences with Ro52 presence between groups. Interestingly, we found that ASSD patients positive for Anti-Jo1 ARS antibody present more important muscle involvement, represented by higher levels of creatine phosphokinase (CPK, *p* = 0.001). Conversely, the most frequent HRCT patterns in the anti-Jo1 group were cryptogenic organized pneumonia (COP, 41.3%) and NSIP (41.3%), while in the non-anti-Jo1 group was NSIP (44.83%). These results are shown in Table 1.

### 3.3. Hardy–Weinberg Equilibrium

Both polymorphisms evaluated meet the HWE (rs8193036, *p*-value = 1, rs2275913, *p*-value = 0.39) for the control group. For this reason, we considered relevant the results observed in both SNPs in the case-control comparison.

### 3.4. Allele and Genotype Frequencies

#### 3.4.1. Case and Control Groups

We did not find statistically significant differences with frequency allele and genotype and dominant and recessive models between case and control groups comparison for the rs8193036 and rs2275913. These results are shown in Table 2.

#### 3.4.2. Anti-Jo1 and Non-Anti-Jo1 Groups

Regarding allele frequencies, we did not find statistically significant differences between the two SNPs evaluated. Interestingly, the non-anti-Jo1 subjects carry the GA genotype of the rs2275913 more frequently than those anti-Jo1 (36.23% vs. 17.25%, respectively), being statistically significative different but not maintained after Bonferroni correction (*p* = 0.03 and *p* =0.06, respectively). In the dominant model, we observed a tendency (*p* = 0.07).

Concerning rs8193036, we found a statistically significant difference with CC genotype between anti-Jo1 and non-anti-Jo1 groups (14.58% vs. 0%, respectively, *p* < 0.001). This finding is maintained in the recessive model and after Bonferroni correction (*p* = 0.002). These results are shown in Table 3.

#### 3.4.3. Anti-Jo1 and HS Groups

When comparing anti-Jo1 patients and HS (Table 4), we only found a tendency with rs2275913 AA genotype (*p* = 0.07, OR = 4.18, 95% CI = 0.97–18.09). Conversely, we found a statistically significant association with rs8193036 CC genotype (*p* = 0.018, OR = 2.91, CI 95% 1.15–7.35) and it is maintained after Bonferroni correction (p= 0.036). In addition, this finding is also maintained in the recessive model (*p* = 0.018, OR = 2.91, CI 95% 1.15–7.35).

#### 3.4.4. Anti-Ro52+ versus Anti-Ro52- and HS Groups

Additionally, we compared the ASSD patients who were positive for the Ro52+ antibody versus those negatives and the HS group. In both comparisons, we did not find significant differences between allele and genotype frequencies with both SNPs. These results are shown in Appendix A. 

### 3.5. Linkage Disequilibrium (LD) and Haplotype Analysis

The haplotype analysis was carried out to determine its association with ASSD susceptibility and the LD between SNPs located in the same gene. This analysis included two SNPs in the IL17A gene, comparing ASSD versus HS, as well as Anti-Jo1 patients versus non-anti-Jo1 and anti-Jo1 versus HS.

Haplotypes and their frequencies are summarized in Figure 1. Figure 1A–C show that the haplotype shaped by rs8193036 and rs2275913 are not in high linkage disequilibrium with an r^2^ value < 80 (r^2^ = 5, 7, and 4, respectively). Moreover, according to the frequencies, we only found a significant difference between anti-Jo1 vs. non-anti-Jo1 patients with CG block (conformed by the minor allele of the rs8193036 and common allele of the rs2275913, *p* = 0.014, OR = 2.35, 95% CI = 1.16–4.77).

## 4. Discussion

The ASSD is a rare and complex autoimmune disease, which displays diverse clinical and serological characteristics. Our results show that anti-Jo1 was the most frequent antibody, arthritis was the most frequent clinical manifestation, and NSIP was the most frequent HRCT pattern, agreeing with previous reports from our research group [7,24]. Although we did not find significant differences between anti-Jo1 and non-anti-Jo1 groups regarding clinical manifestations, we found that arthritis and mechanic’s hands were almost significative more frequent in the anti-Jo1 group, as well as significantly higher CPK serum levels. These findings are similar to those reported by Pinal-Fernandez et al. [25] and Rojas-Serrano et al. [9], who described anti-Jo1+ patients display more arthritis, proximal muscle weakness, Raynaud’s phenomenon, and CPK higher levels than those non-anti-Jo1.

The largest ASSD multicenter cohort until now, shaped by the American and European Network of Antisynthetase Syndrome (AENEAS) collaborative group [26], included 828 patients from 63 hospitals from 10 different countries and described that anti-Jo1 was the most frequent ARS autoantibody, and these patients had more muscle and articular involvement. These findings support the premise that anti-Jo1 ARS could be associated with multi-organ involvement, while non-anti-Jo1 ARS antibodies are mainly limited to lung affection. This hypothesis is supported by Hervier et al. [27] with their cluster analysis between anti-Jo1, anti-PL7, and anti-PL12 ASSD patients, finding that the phenotype and the survival were correlated with the anti-ARS specificity. Furthermore, the most frequent tomographic pattern in anti-Jo1 patients was COP and NSIP in a similar percentage, while in non-anti-Jo1 patients was NSIP. These two tomographic patterns have been widely described as the most prevalent in the ASSD [9,28]. Conversely, Jensen and coworkers [29] showed that NSIP was the most frequent tomographic pattern in their cohort. This finding could be due to differences in the sample size, which is smaller than ours.

On the other hand, previous studies have tried to identify genetic susceptibility for ASSD or IIM-related ILD with inconclusive results. Our research group recently described that the rs1143634/GG genotype of the *IL1B* gene is associated with a higher risk for ASSD in a Mexican mestizo population [30]. Sugiura et al. [31] showed that *STAT4* rs7574865 is associated with DM/PM, as well as ILD-related myopathies in a Japanese population, suggesting that DM/PM with or without ILD shares a common gene associated. In a Chinese population, Chen and coworkers [30] described two SNPs rs7117932 and rs6590330 in the *ETS1*, and the rs951005 in *CCL21* might confer genetic IIM predisposition, as well as IIM-ILD. Conversely, *ANKRD55* SNP rs7731626 was a protective factor for DM/PM-ILD in the Chinese Han population [32]. Recently, López-Matías et al. [33] studied a promoter polymorphism in *MUC5B* (rs35705950) in ASSD, a widely described gene associated with idiopathic pulmonary fibrosis (IPF) [34] and RA-ILD [35], two of the most critical ILDs. However, they did not find an association of rs35705950 with ASSD, suggesting a different pathogenic pathway between these diseases.

For the first time, we described that the rs8193036 CC genotype in *IL17A* is associated with ASSD patients positive for the anti-Jo1 antibody. Also, anti-Jo1 patients display different genotype frequencies compared with non-anti-Jo1 patients. This finding is also replicated in haplotype analysis, where CG block (shaped by minor allele [C] of the rs8193036 and common allele [G] of the rs2275913) is more frequent in anti-Jo1 ASSD patients. This finding could be related to a significant homogenous group of ASSD patients, suggesting a diverse genetic background between subgroups of ASSD according to the various anti-tRNA autoantibodies. Previous studies have established that in SLE, IL-17A provides an essential stimulus for B cells and contributes to the disease’s abnormal autoantibodies profiles [36,37]. As suggested in mice, IL-17A could participate in autoantibody production by forming ectopic lymphoid structures that function as germinal centers [38]. This critical mechanism could be participating in the different serological spectrum in ASSD, where the genetic background could play a key role.

It has been demonstrated that both functional promoter SNPs in *IL17A* could alter IL-17A serum levels, promoting a higher affinity for transcriptional factors [39,40]. Similarly, it has been previously demonstrated that patients with ASSD have increased serum levels of the IL-17A compared with healthy controls. In addition, the patients with refractory ASSD treated with rituximab showed a reduction of the serum levels of the IL-17A [41]. Moreover, our research group described that in ASSD, patients who present ILD progression had higher levels of Th17 related cytokines (IL-17A, IL-6, IL-22) [4], suggesting a potential key role of IL-17A (Th17 CD4+ T cells) in the pathogenesis of ASSD. Supporting this idea, several studies in murine models where a pro-fibrotic effect of IL-17A in the lungs through Smad2/3-STAT3-TGFβ pathway, promoting the proliferation of mesenchymal cells [42,43]. Besides, significantly higher frequency of circulating, skin, and lung infiltrating Th17 cells and higher levels of serum, skin, and lung IL-17A, TGF-β1, IL-6, and RORγt were detected in mice in a bleomycin-induced murine model of systemic sclerosis [44].

Additionally, we carried out an analysis with ASSD patients positive to the anti-Ro52 antibody, compared with those negatives and HS, due to the critical prevalence of this antibody in several autoimmune diseases, including ASSD. We did not find an association between the presence of Ro52 with both SNPs evaluated. To our knowledge, this is the first report where a relationship between anti-Ro52+ ASSD patients and single nucleotide variants in *IL17A* has been investigated. Nonetheless, the relationship between anti-Ro52 and ILD in autoimmune diseases has been reported in several studies, and ASSD is not the exception. Huang et al. [8] and Wu and coworkers [45] described that Ro52 and anti-Jo1 antibodies participate as independent risk factors for IIM-related ILD.

This study is not exempt from limitations. Firstly, we only were able to evaluate two SNPs in the *IL17A* gene. Secondly, we did not include a control group of subjects with IIM without ILD since our center only attends to those with pulmonary involvement. Finally, we did not measure serum levels of IL-17A in subjects included in the study.

## 5. Conclusions

We described for the first time that the rs8193036 CC genotype in *IL17A* is associated with ASSD patients with anti-Jo1 ARS. Also, anti-Jo1 and non-anti-Jo1 patients display differences in genotype and haplotype frequencies. These findings support the hypothesis that could be different genetic component between subgroups of ASSD patients according to ARS antibodies. However, more studies are required to elucidate the role of genetic variants in pro-inflammatory genes.

## Figures and Tables

**Figure 1 life-11-00174-f001:**
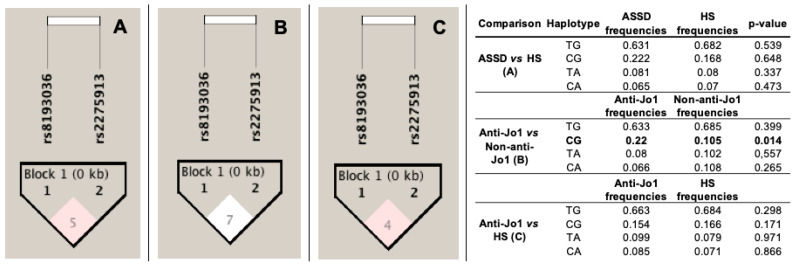
Haplotype analysis. (**A**) Haplotype frequencies between ASSD *versus* HS groups. (**B**) Haplotype frequencies between Anti-Jo1 *versus* Non-anti-Jo1 subgroups. (**C**) Haplotype frequencies between Anti-Jo1 *versus* HS groups. ASSD: antisynthetase syndrome; HS: healthy subjects; Anti-Jo1: ASSD patients positive to Jo1 antibody; non-anti-Jo1: ASSD patients negative to Jo1 antibody. The *p*-value < 0.05 was considered significant.

**Table 1 life-11-00174-t001:** Demographic and clinical variables from antisynthetase syndrome (ASSD) and healthy subjects (HS) groups, and among ASSD anti-Jo1 and non-anti-Jo1.

Variables	ASSD	HS	*p*-Value	Anti-Jo1	Non-Anti-Jo1	*p*-Value
	(*n* = 121)	(*n* = 346)		(*n* = 52)	(*n* = 69)	
Age, (years)	55 (27–83)	55 (21–80)	0.66	54 (41–73)	58 (38–75)	0.29
Sex, female (%)	82 (67.77)	263 (76.01)	0.06	35 (67.31)	47 (79.66)	0.14
BMI, kg/m^2^	27.93 (15.61–51)	27.78 (17.09–45.12)	0.68	27.42 (23–34.29)	27.75 (15.61–34.92)	0.56
Pulmonary function						
FVC, %	60.5 (32–114)			56.5 (32–114)	61 (35–109)	0.43
DLCO, %	50 (2–110)			48.5 (6.25–102)	53 (2–110)	0.85
Arthritis	*n* = 105			*n* = 46	*n* = 59	
Yes (%)	77 (73.33)			38 (82.61)	39 (66.10)	0.057
Mechanic’s hands				*n* = 46	*n* = 59	
Yes (%)	64 (60.95)			30 (65.22)	28 (47.46)	0.07
Fever				*n* = 46	*n* = 59	
Yes (%)	60 (57.14)			30 (65.22)	30 (50.85)	0.14
Raynaud’s phenomenon				*n* = 46	*n* = 59	
Yes (%)	50 (47.62)			24 (52.17)	26 (44.07)	0.41
CPK, U/I	109 (18–14270)			242.5 (24–7210)	67.5 (18–14270)	0.001
Autoantibodies						
Anti-Jo1 (%)	52 (42.98)			52 (100)	0	
Anti-PL12 (%)	39 (32.23)			4 (7.69)	35 (50.72)	
Anti-PL7 (%)	24 (19.83)			3 (5.77)	21 (30.43)	
Anti-EJ (%)	14 (11.57)			1 (1.92)	13 (18.84)	
Anti-OJ (%)	10 (8.26)			0	10 (14.49)	
Anti-Ro52 (%)	62 (51.23)			28 (53.85)	34 (49.28)	0.61
HRCT	*n* = 104			*n* = 46	*n* = 58	
NSIP (%)	45 (43.27)			19 (41.30)	26 (44.83)	0.71
COP (%)	38 (36.54)			19 (41.30)	19 (32.76)	0.37
UIP (%)	17 (16.35)			6 (13.05)	11 (18.97)	0.41
LIP (%)	4 (3.84)			2 (4.35)	2 (3.44)	0.81

ASSD: antisynthetase syndrome; BMI: body mass index; CPK: creatine phosphokinase; DLCO: single-breath carbon monoxide diffusing capacity; FVC: forced vital capacity; NSIP: non-specific interstitial pneumonia; COP: cryptogenic organized pneumonia; UIP: usual interstitial pneumonia; LIP: lymphoid interstitial pneumonia; HRCT: high resolution computed tomography. All values are expressed as median and minimum-maximum values. We used the Mann–Whitney U test and Fisher exact test. *p*-value < 0.05 was considered as significative.

**Table 2 life-11-00174-t002:** Allele and genotype frequencies and genetic models of IL17A single nucleotide polymorphisms (SNPs) in ASSD and HS comparison.

Model	ASSD		HS		*p*-Value	OR	95% CI
	*n* = 120	F (%)	*n* = 340	F (%)			
**rs2275913**							
**Genotypes**							
GG	81	67.50	243	71.47	0.41	0.83	0.53–1.30
GA	34	28.33	92	27.06	0.79	1.07	0.67–1.69
AA	5	4.17	5	1.47	0.08	2.91	0.82–10.24
**Alleles**							
G	196	81.67	578	85	0.22	0.78	0.53–1.16
A	44	18.33	102	15		1.27	0.86–1.88
**Dominant**							
GG	81	67.50	243	71.47	0.41	0.83	0.53–1.30
GA+AA	39	32.50	97	28.53		1.21	0.77–1.89
**Recessive**							
GG+GA	115	95.83	335	98.53	0.08	0.34	0.10–1.21
AA	5	4.17	5	1.47		2.91	0.82–10.24
**rs8193036**							
**Genotypes**	*n* = 115	F (%)	*n* = 343	F (%)			
TT	66	57.39	199	58.02	0.91	0.97	0.64–1.49
TC	42	36.52	125	36.44	0.99	1	0.65–1.56
CC	7	6.09	19	5.54	0.83	1.11	0.45–2.70
**Alleles**							
T	174	75.65	523	76.24	0.86	0.97	0.68–1.37
C	56	24.35	163	23.76		1.03	0.73–1.46
**Dominant**							
TT	66	57.39	199	58.02	0.91	0.97	0.64–1.49
TC+CC	49	42.61	144	41.98		1.03	0.67–1.57
**Recessive**							
TT+TC	108	93.91	324	94.46	0.83	0.90	0.37–2.21
CC	7	6.09	19	5.54		1.11	0.45–2.70

ASSD: antisynthetase syndrome; HS: healthy subjects. *p*-value <0.05 was considered significant.

**Table 3 life-11-00174-t003:** Allele and genotype frequencies of IL17A SNPs in anti-Jo1 and non-anti-Jo1 ASSD patients.

Model	Anti-Jo1		Non-Anti-Jo1		*p*-Value
	*n* = 51	F (%)	*n* = 69	F (%)	
**rs2275913**					
**Genotypes**					
GG	39	76.47	42	60.87	0.07
GA	9	17.65	25	36.23	0.03
AA	3	5.88	2	2.90	0.42
**Alleles**					
G	87	85.29	109	78.99	0.21
A	15	14.71	29	21.01	
**rs8193036**					
**Genotypes**	*n* = 48	F (%)	*n* = 67	F (%)	
TT	27	56.25	39	58.21	0.83
TC	14	29.17	28	41.79	0.17
CC	7	14.58	0	0.00	<0.001
**Alleles**					
T	68	70.83	106	79.10	0.15
C	28	29.17	28	20.90	

Anti-Jo1: ASSD patients positive to Jo1 antibody; non-anti-Jo1: ASSD patients negative to Jo1 antibody; NA: not apply. *p*-value < 0.05 was considered significant and was corrected by the Bonferroni test.

**Table 4 life-11-00174-t004:** Allele and genotype frequencies and genetic models of IL17A SNPs evaluated in Anti-Jo1 ASSD patients and HS.

Model	Anti-Jo1		HS		*p*-Value	*p*-Value Adj-Bon	OR	95% CI
	*n* = 51	F (%)	*n* = 340	F (%)				
**rs2275913**								
**Genotypes**								
GG	39	76.47	243	71.47	0.46		1.30	0.65–2.58
GA	9	17.65	92	27.06	0.15		0.58	0.27–1.23
AA	3	5.88	5	1.47	0.07		4.18	0.97–18.09
**Alleles**								
G	87	85.29	578	85.00	0.94		1.02	0.57–1.84
A	15	14.71	102	15.00			0.98	0.54–1.76
**Dominant**								
GG	39	76.47	243	71.47	0.46		1.30	0.65–2.58
GA+AA	12	23.53	97	28.53			0.77	0.39–1.53
**Recessive**								
GG+GA	48	94.12	335	98.53	0.07		0.24	0.06–1.03
AA	3	5.88	5	1.47			4.18	0.97–18.09
**rs8193036**								
**Genotypes**	*n* = 48	F (%)	*n* = 343	F (%)				
TT	27	56.25	199	58.02	0.82		0.93	0.51–1.71
TC	14	29.17	125	36.44	0.32		0.72	0.37–1.39
CC	7	14.58	19	5.54	0.018	0.036	2.91	1.15–7.35
**Alleles**								
T	68	70.83	523	76.24	0.25		0.76	0.47–1.22
C	28	29.17	163	23.76			1.32	0.82–2.12
**Dominant**								
TT	27	56.25	199	58.02	0.82		0.93	0.51–1.71
TC+CC	21	43.75	144	41.98			1.07	0.58–1.98
**Recessive**								
TT+TC	41	85.42	324	94.46	0.018	0.036	0.16	0.06–0.46
CC	7	14.58	19	5.54			2.91	1.15–7.35

Anti-Jo1: ASSD patients positive to Jo1 antibody; HS: healthy subjects. The *p*-value < 0.05 was considered significant and was corrected by the Bonferroni test.

## Data Availability

All data generated for this study are included in this article and its Appendix A.

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
