# Peer review of "Single Nucleotide Polymorphism in the IL17A Gene Is Associated with Interstitial Lung Disease Positive to Anti-Jo1 Antisynthetase Autoantibodies"

_life, 2021, doi:10.3390/life11020174_

Round 1

Reviewer 1 Report

Dear Authors,

In the manuscript entitled "Single nucleotide polymorphism in the IL17A gene is associated with interstitial lung disease positive to anti-Jo1 antisynthetase autoantibodies" authors described the association of a particular SNP with a group of anti-Jo1 ASSD subjects. 

The manuscript is well written and scientifically relevant. Although the authors stated their limitations, I consider the results consistent and of scientific interest. However, considering the rarity of the disease I think that those results could seriously be helpful in the knowledge of this disease. 

For these reasons, I do recommend the following manuscript for the publication without any major/minor revision.

Best Regards.

Author Response

Thank you very much for your kind comments and for recommending our manuscript without any changes.

Reviewer 2 Report

2 SNPs in the IL17A gene were tested in 121 anti-synthetase syndrome patients with lung disease and 340 healthy controls. The patient group was divided into Jo-1 positive and Jo-1 negative patients and Ro positive and Ro negative patients. The rs8193036/CC genotype in IL17A was significantly associated with Jo-1 positive anti-synthetase syndrome patients with ILD. A similar association has also been noted in patients with RA and SLE.

Major comments:

1) It is not clear as to how the healthy controls were recruited. They appear to be age-matched with the patients but there were more women in the control group. Please provide more detail.

2) As stated in the paper, it is a limitation that no patients without ILD were included. This has led to a potential bias.

Minor

1) Please clarify if the ranges or interquartile ranges are quoted in table 1 - I think it is the former and I wonder if the latter would be more informative.

2) There are some typos e.g. I think it is conventional to use  Jo-1 and mechanics hands. Arthritis should be corrected in table 1.

3) Some of the English phrasing is a little odd.

Author Response

1) It is not clear as to how the healthy controls were recruited. They appear to be age-matched with the patients but there were more women in the control group. Please provide more detail.

R= Thank you. Now we have included a brief sentence describing how we choose the control subjects.

2) As stated in the paper, it is a limitation that no patients without ILD were included. This has led to a potential bias.

Thank you for your observation. As you said, this could be an important bias. However, as we stated in the limitations paragraph, our center only attends lung involvement patients. However, we do not discard possible associations with other research groups to complete more extensive studies in ASSD in the future.

Minor

1) Please clarify if the ranges or interquartile ranges are quoted in table 1 - I think it is the former and I wonder if the latter would be more informative.

R= Thank you. As we stated in the footnote of Table 1, values are expressed in minimum and maximum values. We decided to describe it this way due to our variables follow a non-parametric distribution.

2) There are some typos e.g. I think it is conventional to use Jo-1 and mechanics hands. Arthritis should be corrected in Table 1.

R= Thank you. Now we have corrected the typos.

3) Some of the English phrasing is a little odd.

R= Thank you. Now we have reviewed and changed the incorrect phrases.

Reviewer 3 Report

This manuscript focuses on the analysis of SNPs in the IL17A gene related to ASSD. The study is well performed and the topic is very interesting; however, there are several major points to be addressed:

1) please prepare graphical abstract that will appear alongside the text abstract in the Table of Contents. Please remember that it should summarize the content, but also represent the topic of the article in an attention-grabbing way

2) according to instructions for authors, the abstract should be structured but without headings. Please remove the headings

3) in Table 1 please provide data regarding the presence of all autoantibodies in ASSD groups (anti-Jo1 and non-anti-Jo1)

4) please provide some immunoblots confirming the presence of autoantibodies

5) also, please provide the exact protocol for immunoblotting

6) I truly believe, that table with  standard results of morphology and biochemistry of blood will nicely supplement the study

7) the discussion is very nicely written

Author Response

1) please prepare graphical abstract that will appear alongside the text abstract in the Table of Contents. Please remember that it should summarize the content, but also represent the topic of the article in an attention-grabbing way

R= Thank you. A graphical abstract has been included.

2) according to instructions for authors, the abstract should be structured but without headings. Please remove the headings

R= Thank you. Now we have deleted the headings in the abstract.

3) in Table 1 please provide data regarding the presence of all autoantibodies in ASSD groups (anti-Jo1 and non-anti-Jo1)

R= Thank you, done.

4) please provide some immunoblots confirming the presence of autoantibodies

R= Thank you for your observation. Now we have included an image in the Supplementary material with two immunoblots.

5) also, please provide the exact protocol for immunoblotting

R= Thank you. The clinical laboratory performs the immunoblots analysis, and we only have access to the results. For that reason, we do not have access to the exact protocol; however, in the Supplementary material, we have included a test principle description for the technique.

Reviewer 4 Report

Dear authors,

The results are extensively studied and well presented. I would suggest to omit two paragraphs (lines 259-272 and 286-296) because they they appear as a deviation of the topic.

Author Response

R= Thank you very much for your kind comments. Now we have deleted the second paragraph. We think that the first one is according to the manuscript's context because we described previous studies in other genes associated with interstitial lung disease in inflammatory myopathies and ASSD.

Round 2

Reviewer 3 Report

The manuscript have been revised and, in my opinion, should be published.